# DOMAIN-INDEPENDENT DOMINANCE OF ADAPTIVE METHODS

## ABSTRACT

From a simplified analysis of adaptive methods, we derive AvaGrad, a new optimizer which outperforms SGD on vision tasks when its adaptability is properly tuned. We observe that the power of our method is partially explained by a decoupling of learning rate and adaptability, greatly simplifying hyperparameter search. In light of this observation, we demonstrate that, against conventional wisdom, Adam can also outperform SGD on vision tasks, as long as the coupling between its learning rate and adaptability is taken into account. In practice, AvaGrad matches the best results, as measured by generalization accuracy, delivered by any existing optimizer (SGD or adaptive) across image classification (CIFAR, ImageNet) and character-level language modelling (Penn Treebank) tasks. This later observation, alongside of AvaGrad's decoupling of hyperparameters, could make it the preferred optimizer for deep learning, replacing both SGD and Adam.

## 1 INTRODUCTION

Deep network architectures are becoming increasingly complex, often containing parameters that can be grouped according to multiple functionalities, such as gating, attention, convolution, and generation. Such parameter groups should arguably be treated differently during training, as their gradient statistics might be highly distinct. Adaptive gradient methods designate parameter-wise learning rates based on gradient histories, treating such parameters groups differently and, in principle, promise to be better suited for training complex neural network architectures.

Nonetheless, advances in neural architectures have not been matched by progress in adaptive gradient descent algorithms. SGD is still prevalent, in spite of the development of seemingly more sophisticated adaptive alternatives, such as RMSProp (Dauphin et al., 2015) and Adam (Kingma & Ba, 2015). Such adaptive methods have been observed to yield poor generalization compared to SGD in classification tasks (Wilson et al., 2017), and hence have been mostly adopted for training complex models (Vaswani et al., 2017; Arjovsky et al., 2017). For relatively simple architectures, such as ResNets (He et al., 2016a) and DenseNets (Huang et al., 2017), SGD is still the dominant choice.

At a theoretical level, concerns have also emerged about the current crop of adaptive methods. Recently, Reddi et al. (2018) has identified cases, even in the stochastic convex setting, where Adam (Kingma & Ba, 2015) fails to converge. Modifications to Adam that provide convergence guarantees have been formulated, but have shortcomings. AMSGrad (Reddi et al., 2018) requires non-increasing learning rates, while AdamNC (Reddi et al., 2018) and AdaBound (Luo et al., 2019) require that adaptivity be gradually eliminated during training. Moreover, while most of the recently proposed variants do not provide formal guarantees for non-convex problems, the few current convergence rate analyses in the literature (Zaheer et al., 2018; Chen et al., 2019) do not match SGD's. Section 3 fully details the convergence rates of the most popular Adam variants, along with their shortcomings.

Our contribution is marked improvements to adaptive optimizers, from both theoretical and practical perspectives. At the theoretical level, we focus on convergence guarantees, deriving new algorithms:

- **Delayed Adam.** Inspired by Zaheer et al. (2018)'s analysis of Adam, Section 4 proposes a simple modification for adaptive gradient methods which yields a provable convergence rate of $O(1/\sqrt{T})$ in the stochastic non-convex setting – the same as SGD. Our modification can be implemented by *swapping two lines of code* and preserves adaptivity without incurring extra memory costs. To illustrate these results, we present a non-convex problem where Adam fails to converge to

a stationary point, while Delayed Adam – Adam with our proposed modification – provably converges with a rate of $O(1/\sqrt{T})$.

- **AvaGrad.** Inspecting the convergence rate of Delayed Adam, we show that it would improve with an adaptive global learning rate, which self-regulates based on global statistics of the gradient second moments. Following this insight, Section 5 proposes a new adaptive method, AvaGrad, whose hyperparameters decouple learning rate and adaptability.

Through extensive experiments, Section 6 demonstrates that AvaGrad is not merely a theoretical exercise. AvaGrad performs as well as both SGD and Adam in their respectively favored usage scenarios. Along this experimental journey, we happen to disprove some conventional wisdom, finding adaptive optimizers, including Adam, to be superior to SGD for training CNNs. The caveat is that, excepting AvaGrad, these methods are sensitive to hyperparameter values. AvaGrad is a uniquely attractive adaptive optimizer, yielding near best results over a wide range of hyperparameters.

## 2 PRELIMINARIES

### 2.1 NOTATION

For vectors $a = [a_1, a_2, \dots], b = [b_1, b_2, \dots] \in \mathbb{R}^d$, we use the following notation: $\frac{1}{a}$ for element-wise division ($\frac{1}{a} = [\frac{1}{a_1}, \frac{1}{a_2}, \dots]$), $\sqrt{a}$ for element-wise square root ($\sqrt{a} = [\sqrt{a_1}, \sqrt{a_2}, \dots]$), $a + b$ for element-wise addition ($a + b = [a_1 + b_1, a_2 + b_2, \dots]$), $a \odot b$ for element-wise multiplication ($a \odot b = [a_1 b_1, a_2 b_2, \dots]$). Moreover, $\|a\|$ is used to denote the $\ell_2$-norm: other norms will be specified whenever used (*e.g.,* $\|a\|_\infty$).

For subscripts and vector indexing, we adopt the following convention: the subscript $t$ is used to denote an object related to the $t$-th iteration of an algorithm (*e.g.,* $w_t \in \mathbb{R}^d$ denotes the iterate at time step $t$); the subscript $i$ is used for indexing: $w_i \in \mathbb{R}$ denotes the $i$-th coordinate of $w \in \mathbb{R}^d$. When used together, $t$ precedes $i$: $w_{t,i} \in \mathbb{R}$ denotes the $i$-th coordinate of $w_t \in \mathbb{R}^d$.

### 2.2 STOCHASTIC NON-CONVEX OPTIMIZATION

In the stochastic non-convex setting, we are concerned with the optimization problem:

$$\min_{w \in \mathbb{R}^d} f(w) = \mathbb{E}_{s \sim \mathcal{D}} \left[ f_s(w) \right] \tag{1}$$

where $\mathcal{D}$ is a probability distribution over a set $\mathcal{S}$ of "data points". We also assume that $f$ is $M$-smooth in $w$, as is typically done in non-convex optimization:

$$\forall \, w, w' \; f(w') \leq f(w) + \langle \nabla f(w), w' - w \rangle + \frac{M}{2} \|w - w'\|^2 \tag{2}$$

Methods for stochastic non-convex optimization are evaluated in terms of number of iterations or gradient evaluations required to achieve small loss gradients. This differs from the stochastic convex setting where convergence is measured w.r.t. suboptimality $f(w) - \min_{w \in \mathbb{R}^d} f(w)$. We assume that the algorithm takes a sequence of data points $S = (s_1, \dots, s_T)$ from which it deterministically computes a sequence of parameter settings $w_1, \dots, w_T$ together with a distribution $\mathcal{P}$ over $\{1, \dots, T\}$. We say an algorithm has a convergence rate of $O(g(T))$ if $\mathbb{E}_{\substack{S \sim \mathcal{D}^T \\ t \sim \mathcal{P}(t|S)}} \left[ \|\nabla f(w_t)\|^2 \right] \leq O(g(T))$ where, as defined above, $f(w) = \mathbb{E}_{s \sim \mathcal{D}} \left[ f_s(w) \right]$.

We also assume that the functions $f_s$ have bounded gradients: there exists some $G_\infty$ such that $\|\nabla f_s(w)\|_\infty \leq G_\infty$ for all $s \in \mathcal{S}$ and $w \in \mathbb{R}^d$. Throughout the paper, we also let $G_2$ denote an upper bound on $\|\nabla f_s(w)\|$.

## 3 RELATED WORK

Here we present a brief overview of optimization methods commonly used for training neural networks, along with their convergence rate guarantees for stochastic smooth non-convex problems.

We consider methods which, at each iteration $t$, receive or compute a gradient estimate:

$$g_t := \nabla f_{s_t}(w_t), \quad s_t \sim \mathcal{D} \tag{3}$$

and perform an update of the form:

$$w_{t+1} = w_t - \alpha_t \cdot \eta_t \odot m_t \tag{4}$$

where $\alpha_t \in \mathbb{R}$ is the **global learning rate**, $\eta_t \in \mathbb{R}^d$ are the **parameter-wise learning rates**, and $m_t \in \mathbb{R}^d$ is the update direction, typically defined as:

$$m_t = \beta_{1,t} m_{t-1} + (1 - \beta_{1,t}) g_t \quad \text{and} \quad m_0 = 0. \tag{5}$$

Non-momentum methods such as SGD, AdaGrad, and RMSProp (Dauphin et al., 2015; Duchi et al., 2011) have $m_t = g_t$ (*i.e.,* $\beta_{1,t} = 0$), while momentum SGD and Adam (Kingma & Ba, 2015) have $\beta_{1,t} \in (0,1)$. Note that while $\alpha_t$ can always be absorbed into $\eta_t$, representing the update in this form will be convenient throughout the paper.

SGD uses the same learning rate for all parameters, *i.e.,* $\eta_t = \vec{1}$. Although SGD is simple and offers no adaptation, it has a convergence rate of $O(1/\sqrt{T})$ with either constant, increasing, or decreasing learning rates (Ghadimi & Lan, 2013), and is widely used when training deep networks, especially CNNs (He et al., 2016a; Huang et al., 2017). At the heart of its convergence proof is the fact that $\mathbb{E}_{s_t}[\alpha_t \cdot \eta_t \odot g_t] = \alpha_t \cdot \nabla f(w_t)$.

Popular adaptive methods such as RMSProp (Dauphin et al., 2015), AdaGrad (Duchi et al., 2011), and Adam (Kingma & Ba, 2015) have $\eta_t = \frac{1}{\sqrt{v_t} + \epsilon}$, where $v_t \in \mathbb{R}^d$ is given by:

$$v_t = \beta_{2,t} v_{t-1} + (1 - \beta_{2,t}) g_t^2 \quad \text{and} \quad v_0 = 0. \tag{6}$$

As $v_t$ is an estimate of the second moments of the gradients, the optimizer designates smaller learning rates for parameters with larger uncertainty in their stochastic gradients. However, in this setting $\eta_t$ and $s_t$ are no longer independent, hence $\mathbb{E}_{s_t}[\alpha_t \cdot \eta_t \odot g_t] \neq \alpha_t \cdot \mathbb{E}_{s_t}[\eta_t] \odot \nabla f(w_t)$. This "bias" can cause RMSProp and Adam to present convergence issues, even in the stochastic convex setting (Reddi et al., 2018).

Recently, Zaheer et al. (2018) showed that, with a constant learning rate, RMSProp and Adam have a convergence rate of $O(\sigma^2 + 1/T)$, where $\sigma^2 = \sup_{w \in \mathbb{R}^d} \mathbb{E}_{s \sim \mathcal{D}} \left[ \| \nabla f_s(w) - \nabla f(w) \|^2 \right]$, hence their result does not generally guarantee convergence. Chen et al. (2019) showed that AdaGrad and AMSGrad enjoy a convergence rate of $O(\log T/\sqrt{T})$ when a decaying learning rate is used. Note that both methods constrain $\eta_t$ in some form, the former with $\beta_{2,t} = 1 - 1/t$ (adaptability diminishes with $t$), and the latter explicitly enforces $v_t \geq v_j$ for all $j < t$ ($\eta_t$ is point-wise non-increasing). In both cases, the method is less adaptive than Adam, and yet analyses so far have not delivered a convergence rate that matches SGD's.

## 4 SGD-LIKE CONVERGENCE WITHOUT CONSTRAINED RATES

We first take a step back to note the following: to show that Adam might not converge in the stochastic convex setting, Reddi et al. (2018) provide a stochastic linear problem where Adam fails to converge w.r.t. suboptimality. Since non-convex optimization is evaluated w.r.t. norm of the gradients, a different instance is required to characterize Adam's behavior in this setting.

The following result shows that even for a quadratic problem, Adam indeed does not converge to a stationary point:

**Theorem 1.** *For any $\epsilon \geq 0$ and constant $\beta_{2,t} = \beta_2 \in [0,1)$, there is a stochastic convex optimization problem for which Adam does not converge to a stationary point.*

---

**Algorithm 1** DELAYED ADAM

**Input:** $w_1 \in \mathbb{R}^d, \alpha_t, \epsilon > 0, \beta_{1,t}, \beta_{2,t} \in [0,1)$
1: Set $m_0 = 0, v_0 = 0$
2: **for** $t = 1$ **to** $T$ **do**
3:     Draw $s_t \sim \mathcal{D}$
4:     Compute $g_t = \nabla f_{s_t}(w_t)$
5:     $m_t = \beta_{1,t} m_{t-1} + (1 - \beta_{1,t}) g_t$
6:     $\eta_t = \frac{1}{\sqrt{v_{t-1}} + \epsilon}$
7:     $w_{t+1} = w_t - \alpha_t \cdot \eta_t \odot m_t$
8:     $v_t = \beta_{2,t} v_{t-1} + (1 - \beta_{2,t}) g_t^2$
9: **end for**

---

*Proof.* The full proof is given in Appendix A. The argument follows closely from Reddi et al. (2018), where we explicitly present a stochastic optimization problem:

$$\min_{w\in[0,1]} f(w) := \mathbb{E}_{s\sim\mathcal{D}}\left[f_s(w)\right] \qquad f_s(w) = \begin{cases} C\frac{w^2}{2}, & \text{with probability} \quad p := \frac{1+\delta}{C+1} \\ -w, & \text{otherwise} \end{cases} \tag{7}$$

We show that, for large enough $C$ (as a function of $\delta, \epsilon, \beta_2$), Adam will move towards $w = 1$ where $\nabla f(1) = \delta$, and that the constraint $w \in [0,1]$ does not make $w = 1$ a stationary point. $\qquad\square$

This result, like the one in Reddi et al. (2018), relies on the fact that $\eta_t$ and $s_t$ are correlated: upon a draw of the rare sample $C\frac{w^2}{2}$, the learning rate $\eta_t$ decreases significantly and Adam takes a small step in the correct direction. On the other hand, a sequence of common samples increases $\eta_t$ and Adam moves faster towards $w = 1$.

Instead of enforcing $\eta_t$ to be point-wise non-increasing in $t$ (Reddi et al., 2018), which forces the optimizer to take small steps even for a long sequence of common samples, we propose to simply have $\eta_t$ be independent of $s_t$. As an extra motivation for this approach, note that successful proof strategies (Zaheer et al., 2018) to analyzing adaptive methods include the following step:

$$\mathbb{E}_{s_t}\left[\eta_t \odot g_t\right] = \mathbb{E}_{s_t}\left[(\eta_{t-1} + \eta_t - \eta_{t-1}) \odot g_t\right] = \eta_{t-1} \odot \nabla f(w_t) + \mathbb{E}_{s_t}\left[(\eta_t - \eta_{t-1}) \odot g_t\right] \tag{8}$$

where bounding $\mathbb{E}_{s_t}\left[(\eta_t - \eta_{t-1}) \odot g_t\right]$, seen as a form of bias, is a key part of recent convergence analyses. Replacing $\eta_t$ by $\eta_{t-1}$ in the update equation of Adam removes this bias and can be implemented by simply swapping lines of code (updating $\eta$ *after* $w$), yielding a simple convergence analysis without hindering the adaptability of the method in any way. Algorithm 1 provides pseudocode when applying this modification, highlighted in red, to Adam, yielding Delayed Adam. The following Theorem shows that this modification is enough to guarantee a SGD-like convergence rate of $O(1/\sqrt{T})$ in the stochastic non-convex setting for general adaptive gradient methods.

**Theorem 2.** *Consider any optimization method which updates parameters as follows:*

$$w_{t+1} = w_t - \alpha_t \cdot \eta_t \odot g_t \tag{9}$$

*where $g_t := \nabla f_{s_t}(w_t)$, $s_t \sim \mathcal{D}$, and $\alpha_t, \eta_t$ are independent of $s_t$.*

*Assume that $f(w_1) - f(w^\star) \leq D$, $f(w) = \mathbb{E}_{s\sim\mathcal{D}}\left[f_s(w)\right]$ is $M$-smooth, and $\|\nabla f_s(w)\|_\infty \leq G_\infty$ for all $s \in \mathcal{S}, w \in \mathbb{R}^d$. Moreover, let $Z = \sum_{t=1}^T \alpha_t \min_i \eta_{t,i}$.*

*For $\alpha_t = \gamma_t \sqrt{\frac{2D}{TMG_\infty^2}}$, if $p(Z|s_t) = p(Z)$ for all $s_t \in \mathcal{S}$, then:*

$$\mathbb{E}_{\substack{S\sim\mathcal{D}^T \\ t\sim\mathcal{P}(t|S)}}\left[\|\nabla f(w_t)\|^2\right] \leq \sqrt{\frac{MDG_\infty^2}{2T}} \cdot \mathbb{E}_{S\sim\mathcal{D}^T}\left[\frac{\sum_{t=1}^T 1 + \gamma_t^2 \|\eta_t\|^2}{\sum_{t=1}^T \gamma_t \min_i \eta_{t,i}}\right] \tag{10}$$

*where $\mathcal{P}$ assigns probabilities $p(t) \propto \alpha_t \cdot \min_i \eta_{t,i}$.*

*Proof.* The full proof is given in Appendix B, along with analysis for the case with momentum $\beta_{1,t} \in (0,1)$ in Appendix B.1, and in particular $\beta_{1,t} = \beta_1/\sqrt{t}$, which yields a similar rate. $\qquad\square$

The convergence rate depends on $\|\eta_t\|$ and $\min_i \eta_{t,i}$, which are random variables for Adam-like algorithms. However, if there are constants $H$ and $L$ such that $0 < L \leq \eta_{t,i} \leq H < \infty$ for all $i$ and $t$, then a rate of $O(1/\sqrt{T})$ is guaranteed. This is the case for Delayed Adam, where $1/(G_2 + \epsilon) \leq \eta_{t,i} \leq 1/\epsilon$ for all $t$ and $i$. Theorem 2 also requires that $\alpha_t$ and $\eta_t$ are independent of $s_t$, which can be assured to hold by applying a "delay" to their respective computations, if necessary (*i.e.*, replacing $\eta_t$ by $\eta_{t-1}$, as in Delayed Adam).

Additionally, the assumption that $p(Z|s_t) = p(Z)$, meaning that a single sample should not affect the distribution of $Z = \sum_{t=1}^T \alpha_t \min_i \eta_{t,i}$, is required since $\mathcal{P}$ is conditioned on the samples $S$ (unlike in standard analysis, where $Z = \sum_{t=1}^T \alpha_t$ and $\alpha_t$ is deterministic), and is expected to hold as $T \to \infty$. Practitioners typically use the last iterate $w_T$ or perform early-stopping: in this case, whether the assumption holds or not does not affect the behavior of the algorithm. Nonetheless, we also show in Appendix B.2 a similar rate that does not require this assumption to hold, which also yields a $O(1/\sqrt{T})$ convergence rate taken that the parameter-wise learning rates are bounded from above and below.

## 5 AVAGRAD: AN ADAPTIVE METHOD WITH ADAPTIVE VARIANCE

Now, we consider the implications of Theorem 2 for Delayed Adam, where $\eta_t = \frac{1}{\sqrt{v_{t-1}}+\epsilon}$, and hence $1/(G_2 + \epsilon) \le \eta_{t,i} \le 1/\epsilon$ for all $t$ and $i$.

For a fixed $\gamma_t = \gamma$, chosen a-priori (that is, without knowledge of the realization of $\{\eta_t\}_{t=1}^T$), we can optimize $\gamma$ to minimize the worst-case rate using $\|\eta_t\|^2 \le d/\epsilon^2$ and $\min_i \eta_{t,i} \ge G_2 + \epsilon$. This yields $\gamma^* = O(\epsilon)$, and a convergence rate of $O(1/\epsilon)$, suggesting that, at least in the worst case, $\epsilon$ should be chosen to be as large as possible, and the learning rate $\alpha$ should scale linearly with $\epsilon$ (hence, also being large).

---

**Algorithm 2** AVAGRAD

**Input:** $w_1 \in \mathbb{R}^d, \alpha_t, \epsilon > 0, \beta_{1,t}, \beta_{2,t} \in [0,1)$
1: Set $m_0 = 0, v_0 = 0$
2: **for** $t = 1$ **to** $T$ **do**
3:  Draw $s_t \sim \mathcal{D}$
4:  Compute $g_t = \nabla f_{s_t}(w_t)$
5:  $m_t = \beta_{1,t}m_{t-1} + (1 - \beta_{1,t})g_t$
6:  $\eta_t = \frac{1}{\sqrt{v_{t-1}}+\epsilon}$
7:  $w_{t+1} = w_t - \alpha_t \cdot \frac{\eta_t}{\left\|\eta_t/\sqrt{d}\right\|_2} \odot m_t$
8:  $v_t = \beta_{2,t}v_{t-1} + (1 - \beta_{2,t})g_t^2$
9: **end for**

---

What if we allow $\gamma_t$ to vary in each time step? For example, choosing $\gamma_t = \sqrt{d}/\|\eta_t\|$ yields a convergence rate with a linear dependence on $\left(\frac{1}{T}\sum_{t=1}^T \frac{\min_i \eta_{t,i}}{\|\eta_t\|}\right)^{-1} \le \left(\frac{1}{T\sqrt{d}}\sum_{t=1}^T \frac{\min_i \eta_{t,i}}{\max_i \eta_{t,i}}\right)^{-1}$. While using $1/(G_2 + \epsilon) \le \eta_{t,i} \le 1/\epsilon$ we see that in the worst-case this is also $O(1/\epsilon)$, this dependence differs from the one with fixed $\gamma_t = \gamma$ in a few aspects. Most notably, if we consider different scalings of $\eta_t = \frac{1}{\sqrt{v_{t-1}}+\epsilon}$ (e.g., small $\epsilon$ and varying $v_{t-1}$), the convergence rate with fixed $\gamma$ can get arbitrarily worse, while for $\gamma_t = \sqrt{d}/\|\eta_t\|$ it remains unchanged. In particular, for the case $d = 1$, we have $\left(\frac{1}{T}\sum_{t=1}^T \frac{\min_i \eta_{t,i}}{\|\eta_t\|}\right)^{-1} = 1$, while for fixed $\gamma$ we get a dependence on $\frac{\sum_{t=1}^T 1+\gamma^2\eta_t^2}{\sum_{t=1}^T \gamma\eta_t}$, which again can be large if $\eta_t$ is either large or small. Lastly, multiplying the learning rate by $\sqrt{d}/\|\eta_t\|$ removes its dependence on $\epsilon$ in the worst-case setting, making the two hyperparameters more separable.

The choice of $\gamma_t = \sqrt{d}/\|\eta_t\|$, motivated by the above facts, yields a method which we name AvaGrad – **A**daptive **VA**riance **Grad**ients, presented as pseudo-code in Algorithm 2 with the proposed scaling highlighted in red. We call it an "adaptive variance" method since, if we scale up or down the variance of the gradients, and hence also $v_t$ and $\eta_t$, the convergence guarantee in Theorem 2 does not change, while for a fixed learning rate (as is not uncommonly done in practice, except for discrete decays during training (Zagoruyko & Komodakis, 2016; Merity et al., 2018)) it can get arbitrarily bad.

## 6 EXPERIMENTS

### 6.1 SYNTHETIC DATA

To illustrate empirically the implications of Theorem 1 and Theorem 2, we set up a synthetic stochastic optimization problem with the same form as the one used in the proof of Theorem 1:

$$\min_{w \in [0,1]} f(w) := \mathbb{E}\left[f_s(w)\right] \qquad f_s(w) = \begin{cases} 999\frac{w^2}{2}, & \text{with probability} \quad 0.002 \\ -w, & \text{otherwise} \end{cases} \qquad (11)$$

This function has a stationary point $w^\star = \frac{1-0.002}{999 \cdot 0.002} \approx \frac{1}{2}$, and it satisfies Theorem 1 for $\beta_1 = 0, \beta_2 = 0.99, \epsilon = 10^{-8}$. We proceed to perform stochastic optimization with Adam, AMSGrad, and Delayed Adam, with constant learning rate $\alpha_t = 10^{-5}$. For simplicity, we let $\mathcal{P}$ be uniform over $(1, \ldots, T)$, since $\alpha_t$ is constant.

Figure 1 shows the progress of $\frac{1}{t}\sum_{t'=1}^t w_{t'}$ and $\frac{1}{t}\sum_{t'=1}^t \|\nabla f(w_{t'})\|^2$ for each iteration $t$: as expected, Adam fails to converge to the stationary point $w^\star$, while both AMSGrad and Delayed Adam converge. Note that Delayed Adam converges significantly faster, likely because it has no constraint on the learning rates.

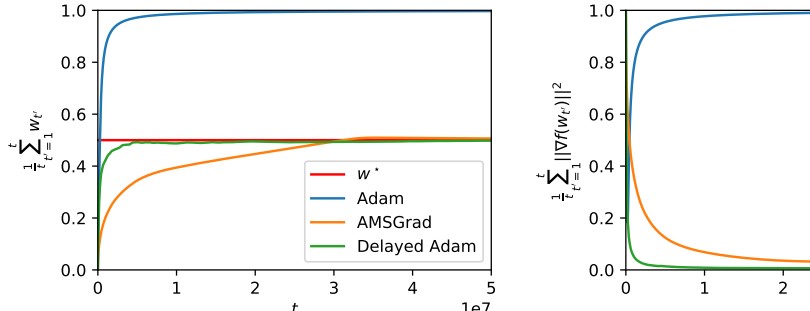

Figure 1: Plots of Adam, AMSGrad, and Delayed Adam trained on the synthetic example in Equation 11, with a stationary point at $w^\star \approx 0.5$. **Left:** The expected iterate sampled uniformly from $\{w_1, \ldots, w_t\}$, for each iteration $t$. As predicted by our theoretical results, Adam moves towards $w = 1$ with $\|\nabla f(w)\| = 1$, while Delayed Adam converges to $w^\star$. **Right:** The expected norm squared of the gradient, for $w$ randomly sampled from $\{w_1, \ldots, w_t\}$. Delayed Adam converges significantly faster than AMSGrad, while Adam fails to converge.

## 6.2 IMAGE CLASSIFICATION WITH CNNS

Our theory suggests that, in the worst case, $\epsilon$ should be chosen as large as possible, at which point the learning rate $\alpha$ should scale linearly with it. As a first experiment to assess this hypothesis, we analyze the interaction between $\alpha$ and $\epsilon$ when training a Wide ResNet 28-4 (Zagoruyko & Komodakis, 2016) on the CIFAR-10 dataset (Krizhevsky, 2009). Our training follows Zagoruyko & Komodakis (2016): images are channel-wise normalized, randomly cropped, and horizontally flipped during training. The learning rate is decayed by a factor of 5 at epochs 60, 120 and 160, and the model is trained for a total of 200 epochs with a weight decay of 0.0005. Appendix C describes additional experimental details.

We use a validation set of 5,000 images to evaluate the performance of SGD and different adaptive gradient methods: Adam, AMSGrad, AdaBound (Luo et al., 2019), AdaShift (Zhou et al., 2019), and our proposed algorithm, AvaGrad. Additionally, we also assess whether performing weight decay as proposed in Loshchilov & Hutter (2019) instead of standard $L_2$ regularization positively impacts the performance of adaptive methods: we do this by also evaluating AdamW and AvaGradW.

We run each adaptive method with different powers of 10 for $\epsilon$, from its default value $\epsilon = 10^{-8}$ up to $\epsilon = 100$, which is large enough such that adaptability should be almost completely removed from the algorithm. We also vary the learning rate $\alpha$ of each method with different powers of 10, multiplied by 1 and 5 (*e.g.,* $0.05, 0.1, 0.5, 1.0, \ldots$). Figure 2 shows the results for Adam and AvaGrad. Our main findings are twofold:

- The optimal $\epsilon$ for every adaptive method is considerably larger than the values typically used in practice, ranging from 0.1 (Adam, AMSGrad, AvaGradW) to 10.0 (AvaGrad, AdamW). For Adam and AMSGrad, the optimal learning rate is $\alpha = \epsilon = 0.1$, a value 100 times larger than the default.

- All adaptive methods, except for AdaBound, outperform SGD in terms of validation performance. Note that for SGD the optimal learning rate is $\alpha = 0.1$, matching the value used in work such as He et al. (2016a); Zagoruyko & Komodakis (2016); Xie et al. (2017), which presented state-of-the-art results at time of publication.

However, the fact that adaptive methods outperform SGD in this setting is not conclusive, since they are executed with more hyperparameter settings (varying $\epsilon$ as well as $\alpha$). Moreover, the main motivation for adaptive methods is to be less sensitive to hyperparameter values; performing an extensive grid search defeats their purpose.

Aiming for a fair comparison between SGD and adaptive methods, we also train a Wide ResNet 28-10 on both CIFAR-10 and CIFAR-100, evaluating the test performance of each adaptive method with its optimal values for $\alpha$ and $\epsilon$ found in the previous experiment. For SGD, we confirmed that

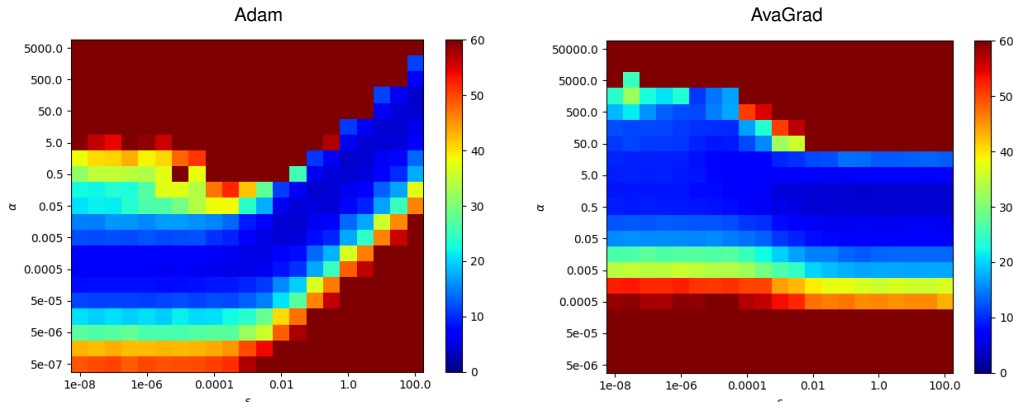

Figure 2: Validation error of a Wide ResNet 28-4 trained on the CIFAR-10 dataset with Adam (**left**) and AvaGrad (**right**), for different values of the learning rate $\alpha$ and parameter $\epsilon$, where larger $\epsilon$ yields less adaptability. Best performance is achieved with small adaptability ($\epsilon > 0.001$).

Table 1: Test performance of SGD and popular adaptive methods in benchmark tasks. Red indicates results with the recommended optimizer, following the paper that proposed each model, and any improved performance is given in blue. The best result for each task is in bold, and numbers in parentheses present standard deviations of 3 runs for CIFAR.

| Method | CIFAR-10 (Test Err %) | CIFAR-100 (Test Err %) | ImageNet (Top-1 Val Err %) | Penn Treebank (Test Bits per Character) |
|---|---|---|---|---|
| SGD | 3.86 (0.08) | 19.05 (0.24) | 24.01 | 1.238 |
| Adam | **3.64 (0.06)** | 18.96 (0.21) | **23.45** | 1.182 |
| AMSGrad | 3.90 (0.17) | 18.97 (0.09) | 23.46 | 1.187 |
| AdaBound | 5.40 (0.24) | 22.76 (0.17) | 27.99 | 2.863 |
| AdaShift | 4.08 (0.11) | 18.88 (0.06) | N/A | 1.274 |
| AdamW | 4.11 (0.17) | 20.13 (0.22) | 27.10 | 1.230 |
| AvaGrad | 3.80 (0.02) | **18.76 (0.20)** | 23.58 | 1.179 |
| AvaGradW | 3.97 (0.02) | 19.04 (0.37) | 23.49 | **1.175** |

the learning rate $\alpha = 0.1$ still yielded the best validation performance with the new architecture, hence the fact that we transfer hyperparameters from the Wide ResNet 28-4 runs does not unfairly advantage adaptive methods in the comparison with SGD. With a larger network and a different task (CIFAR-100), this experiment should also capture how hyperparameters of adaptive methods transfer between tasks and models.

On CIFAR-10, SGD achieves 3.86% test error (reported as 4% in Zagoruyko & Komodakis (2016)) and is outperformed by both Adam (3.64%) and AvaGrad (3.80%). On CIFAR-100, SGD (19.05%) is outperformed by Adam (18.96%), AMSGrad (18.97%), AdaShift (18.88%), AvaGrad (18.76%), and AvaGradW (19.04%). We believe these results are surprising, as they show that adaptive methods can yield state-of-the-art performance when training CNNs as long as their adaptability is correctly controlled with $\epsilon$.

As a final evaluation of the role of adaptability when training convolutional networks, we repeat the previous experiment on the ImageNet dataset (Russakovsky et al., 2015), training a ResNet 50 (He et al., 2016b) with SGD and different adaptive methods, transferring the hyperparameters from our original CIFAR-10 results. Training follows Gross & Wilber (2016): the network is trained for 100 epochs with a batch size of 256, the learning rate is decayed by a factor of 10 at epochs 30, 60 and 90, and a weight decay of 0.0001 is applied. SGD yields 24.01% top-1 validation error, underperforming Adam (23.45%), AMSGrad (23.46%), AvaGrad (23.58%) and AvaGradW (23.49%). Again, note that the hyperparameters used for SGD match the ones in He et al. (2016a), He et al. (2016b) and Gross & Wilber (2016): an initial learning rate of 0.1 with a momentum of 0.9. Table 1 summarizes.

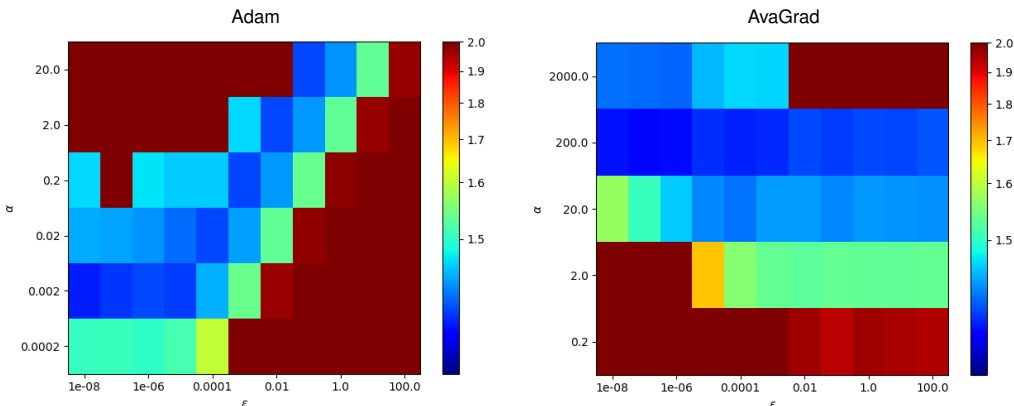

Figure 3: Validation bits-per-character (*lower is better*) of a 3-layer LSTM with 300 hidden units, trained on the Penn Treebank dataset with Adam (**left**) and AvaGrad (**right**), for different values of the learning rate $\alpha$ and parameter $\epsilon$, where larger $\epsilon$ yields less adaptability. Best performance is achieved with high adaptability ($\epsilon < 0.0001$).

## 6.3 LANGUAGE MODELLING WITH RNNS

It is perhaps not very surprising that to perform optimally in the image classification tasks studied previously, adaptive gradient methods required large values of $\epsilon$, and hence were *barely* adaptive. Here, we consider a task where state-of-the-art results are typically achieved by adaptive methods with low values for $\epsilon$: language modelling with recurrent networks. In particular, we perform character-level language modelling on Penn Treebank dataset (Marcus et al., 1994; Mikolov et al., 2010) with LSTMs, following Merity et al. (2018). The model is trained for 500 epochs, and the learning rate is decayed by 10 at epochs 300 and 400. A batch size of 128, a BPTT length of 150, and weight decay of $1.2 \times 10^{-6}$ are used, along with dropout.

We evaluate the validation performance of SGD, Adam, AMSGrad, AdaShift, AdaBound, AdamW, AvaGrad and AvaGradW with varying learning rate $\alpha$ and adaptability parameter $\epsilon$, when training a 3-layer LSTM with 300 hidden units in each layer. Figure 3 shows that, in this task, smaller values for $\epsilon$ are indeed optimal: Adam, AMSGrad and AvaGrad performed best with $\epsilon = 10^{-8}$. The optimal learning rates for both Adam and AMSGrad, $\alpha = 0.002$, agree with the value used in Merity et al. (2018). Both AvaGrad and AvaGradW performed best with $\alpha = 200$: the former with $\epsilon = 10^{-8}$, the latter with $\epsilon = 10^{-5}$.

Next, we train a 3-layer LSTM with 1000 hidden units per layer (the same model used in Merity et al. (2018), where it was trained with Adam), choosing values for $\alpha, \epsilon$ which yielded the best validation performance in the previous experiment. For SGD, we again confirmed that a learning rate of 20 performed best on the validation set. Table 1 (right column) reports all results.

In this setting, AvaGrad and AvaGradW outperform Adam, achieving bit-per-characters of 1.179 and 1.175 compared to 1.182.

## 6.4 HYPERPARAMETER SEPARABILITY AND DOMAIN-INDEPENDENCE

One of the main motivations behind AvaGrad is that it removes the dependence between the learning rate $\alpha$ and the adaptability parameter $\epsilon$, at least in the worst-case rate of Theorem 2. Observing the heatmaps in Figure 2 and 3, we can see that indeed AvaGrad offers more separability between $\alpha$ and $\epsilon$, when compared to Adam. For example, for $\epsilon \geq 0.0001$, it has little to no interaction with the learning rate $\alpha$, as opposed to Adam where the optimal $\alpha$ increases linearly with $\epsilon$. For language modelling on Penn Treebank, the optimal learning rate for AvaGrad was $\alpha = 200$ *for every choice of* $\epsilon$, while for image classification on CIFAR-10, we had $\alpha = 1.0$ for all values of $\epsilon$ except for $\epsilon \in \{10^{-5}, 100\}$. This suggests that AvaGrad enables a grid search over $\alpha$ and $\epsilon$ (with quadratic complexity) to be broken into two line searches over $\alpha$ and $\epsilon$ separately (linear complexity).

## 7 CONCLUSION

As neural architectures become more complex, with parameters having highly heterogeneous roles, parameter-wise learning rates are often necessary for training. However, adaptive methods have both theoretical and empirical gaps, with SGD outperforming them in some tasks and having stronger theoretical convergence guarantees. In this paper, we close this gap, by first providing a convergence rate guarantee that matches SGD's, and by showing that, with proper hyperparameter tuning, adaptive methods can dominate in both computer vision and natural language processing tasks. Key to our finding is AvaGrad, our proposed optimizer whose adaptability is decoupled from its learning rate.

Our experimental results show that proper tuning of the learning rate together with the adaptability of the method is necessary to achieve optimal results in different domains, where distinct neural network architectures are used across tasks. By enabling this tuning to be performed in linear time, AvaGrad takes a leap towards efficient domain-agnostic training of general neural architectures.

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

# Appendix

## A  PROOF OF THEOREM 1

*Proof.* Consider the following stochastic optimization problem:

$$\min_{w \in [0,1]} f(w) := \mathbb{E}_{s \sim \mathcal{D}} \left[ f_s(w) \right] \qquad f_s(w) = \begin{cases} C \frac{w^2}{2}, & \text{with probability} \quad p := \frac{1+\delta}{C+1} \\ -w, & \text{otherwise} \end{cases} \tag{12}$$

where $C > \frac{1-p}{p} > 1 + \frac{\epsilon}{w_1 \sqrt{1-\beta_2}}$. Note that $\nabla f(w) = pCw - (1-p)$, and $f$ is minimized at $w^\star = \frac{1-p}{Cp} = \frac{C-\delta}{C(1+\delta)}$.

The proof follows closely from Reddi's linear example for convergence in suboptimality. We assume w.l.o.g. that $\beta_1 = 0$. Consider:

$$\Delta_t = w_{t+1} - w_t = -\eta \frac{g_t}{\sqrt{v_t} + \epsilon} = -\eta \frac{g_t}{\sqrt{\beta_2 v_{t-1} + (1-\beta_2) g_t^2} + \epsilon} \tag{13}$$

$$\frac{\mathbb{E}\left[\Delta_t\right]}{\eta} = \frac{\mathbb{E}\left[w_{t+1} - w_t\right]}{\eta} = -\mathbb{E}\left[\frac{g_t}{\sqrt{\beta_2 v_{t-1} + (1-\beta_2) g_t^2} + \epsilon}\right]$$

$$= p\mathbb{E}\left[\underbrace{\frac{-Cw_t}{\sqrt{\beta_2 v_{t-1} + (1-\beta_2) C^2 w_t^2} + \epsilon}}_{T_1}\right] + (1-p)\mathbb{E}\left[\underbrace{\frac{1}{\sqrt{\beta_2 v_{t-1} + (1-\beta_2)} + \epsilon}}_{T_2}\right] \tag{14}$$

where the expectation is over all the randomness in the algorithm up to time $t$, as all expectations to follow in the proof. Note that $T_1 = 0$ for $w_t = 0$. For $w_t > 0$ we bound $T_1$ as follows:

$$T_1 \geq \frac{-Cw_t}{\sqrt{(1-\beta_2) C^2 w_t^2}} = \frac{-1}{\sqrt{1-\beta_2}} \tag{15}$$

Hence, $T_1 \geq \min(0, \frac{-1}{\sqrt{1-\beta_2}}) = \frac{-1}{\sqrt{1-\beta_2}}$.

As for $T_2$, we have, from Jensen's inequality:

$$\mathbb{E}\left[T_2\right] \geq \frac{1}{\sqrt{\beta_2 \mathbb{E}\left[v_{t-1}\right] + 1 - \beta_2} + \epsilon} \tag{16}$$

Now, remember that $v_{t-1} = (1-\beta_2) \sum_{i=1}^{t-1} \beta_2^{t-i-1} g_i^2$, hence:

$$\mathbb{E}\left[v_{t-1}\right] = (1-\beta_2) \sum_{i=1}^{t-1} \beta_2^{t-i-1} \mathbb{E}\left[g_i^2\right]$$

$$= (1-\beta_2) \sum_{i=1}^{t-1} \beta_2^{t-i-1} \left(1 - p + pC^2 \mathbb{E}\left[w_t^2\right]\right) \tag{17}$$

$$\leq (1-\beta_2) \sum_{i=1}^{t-1} \beta_2^{t-i-1} \left(1 - p + pC^2\right)$$

$$\leq (1 - \beta_2^{t-1}) \left(1 - p + pC^2\right) \leq (1+\delta)C^2$$

and thus:

$$\mathbb{E}\left[T_2\right] \geq \frac{1}{\sqrt{\beta_2 (1+\delta)C + 1 - \beta_2} + \epsilon} \tag{18}$$

Plugging in the bounds for $T_1$ and $T_2$ in Equation 14:

$$\frac{\mathbb{E}\left[\Delta_t\right]}{\eta} \geq \frac{1+\delta}{C+1} \frac{-1}{\sqrt{1-\beta_2}} + \left(1 - \frac{1+\delta}{C+1}\right) \frac{1}{\sqrt{\beta_2(1+\delta)C + 1 - \beta_2} + \epsilon} \quad (19)$$

Hence, for large enough $C$, and $C \gg \delta$, $w^\star \approx \frac{1}{1+\delta}$ while the above quantity becomes non-negative, and hence $\mathbb{E}\left[w_t\right] \geq w_1$. In other words, Adam will, in expectation, drift away from the stationary point, towards $w = 1$, at which point $\|\nabla f(1)\|^2 = \delta$. For example, $\delta = 1$ implies that $\lim_{T\to\infty} \frac{1}{T} \sum_{t=1}^{T} \mathbb{E}\left[\|\nabla f(w_t)\|^2\right] = 1$. To see that $w = 1$ is not a stationary point due to the feasibility constraints, check that $\nabla f(1) = 1 > 0$: that is, the negative gradient points *towards* the feasible region. $\qquad \square$

## B    PROOF OF THEOREM 2

*Proof.* Throughout the proof we use the following notation for clarity:

$$H_t = \max_i \eta_{t,i} \qquad L_t = \min_i \eta_{t,i} \quad (20)$$

We start from the fact that $f$ is $M$-smooth:

$$f(w_{t+1}) \leq f(w_t) + \langle \nabla f(w_t), w_{t+1} - w_t \rangle + \frac{M}{2} \|w_{t+1} - w_t\|^2 \quad (21)$$

and use the update $w_{t+1} = w_t - \alpha_t \cdot \eta_t \odot m_t$:

$$
\begin{aligned}
f(w_{t+1}) &\leq f(w_t) - \alpha_t \langle \nabla f(w_t), m_t \odot \eta_t \rangle + \frac{\alpha_t^2 M}{2} \|m_t \odot \eta_t\|^2 \\
&\leq f(w_t) - \alpha_t \langle \nabla f(w_t), m_t \odot \eta_t \rangle + \frac{\alpha_t^2 M G_\infty^2 \|\eta_t\|^2}{2} \\
&\leq f(w_t) - \alpha_t \beta_{1,t} \langle \nabla f(w_t), m_{t-1} \odot \eta_t \rangle - \alpha_t(1 - \beta_{1,t}) \langle \nabla f(w_t), g_t \odot \eta_t \rangle + \frac{\alpha_t^2 M G_\infty^2 \|\eta_t\|^2}{2} \\
&\leq f(w_t) + \alpha_t \beta_{1,t} \|\nabla f(w_t)\| \cdot \|m_{t-1} \odot \eta_t\| - \alpha_t(1 - \beta_{1,t}) \langle \nabla f(w_t), g_t \odot \eta_t \rangle + \frac{\alpha_t^2 M G_\infty^2 \|\eta_t\|^2}{2} \\
&\leq f(w_t) + \alpha_t \beta_{1,t} G_2^2 H_t - \alpha_t(1 - \beta_{1,t}) \langle \nabla f(w_t), g_t \odot \eta_t \rangle + \frac{\alpha_t^2 M G_\infty^2 \|\eta_t\|^2}{2}
\end{aligned}
$$
$$\quad (22)$$

where in the first step we used the fact that $\|m_t \odot \eta_t\|^2 = \sum_{i=1}^{d} m_{t,i}^2 \eta_{t,i}^2 \leq \max_j m_{t,j}^2 \sum_{i=1}^{d} \eta_{t,i}^2 \leq G_\infty^2 \|\eta_t\|^2$, in the second we used $m_t = \beta_{1,t} m_{t-1} + (1 - \beta_{1,t}) g_t$, in the third we used Cauchy-Schwarz, and in the fourth we used $\|\nabla f(w_t)\| \leq G_2$, along with $\|m_{t-1} \odot \eta_t\| = \sqrt{\sum_{i=1}^{d} m_{t-1,i}^2 \eta_{t,i}^2} \leq \max_j \eta_{t,j} \sqrt{\sum_{i=1}^{d} m_{t-1,i}^2} \leq G_2 H_t$.

Now, taking the expectation over $s_t$, and using the fact that $\mathbb{E}_{s_t}\left[g_t\right] = \nabla f(w_t)$, and that $\eta_t, \alpha_t$ are both independent of $s_t$:

$$
\begin{aligned}
\mathbb{E}_{s_t}\left[f(w_{t+1})\right] &\leq f(w_t) + \alpha_t \beta_{1,t} G_2^2 H_t - \alpha_t(1 - \beta_{1,t}) \langle \nabla f(w_t), \nabla f(w_t) \odot \eta_t \rangle + \frac{\alpha_t^2 M G_\infty^2 \|\eta_t\|^2}{2} \\
&\leq f(w_t) + \alpha_t \beta_{1,t} G_2^2 H_t - \alpha_t(1 - \beta_1) \|\nabla f(w_t)\|^2 L_t + \frac{\alpha_t^2 M G_\infty^2 \|\eta_t\|^2}{2}
\end{aligned}
$$
$$\quad (23)$$

where in the second step we used $\beta_{1,t} \leq \beta_1$ and $\langle \nabla f(w_t), \nabla f(w_t) \odot \eta_t \rangle = \sum_{i=1}^{d} \nabla f(w)_i^2 \eta_{t,i} \geq \min_j \eta_{t,j} \sum_{i=1}^{d} \nabla f(w)_i^2 = L_t \|\nabla f(w)\|^2$.

Re-arranging, we get:

$$\alpha_t L_t(1 - \beta_1) \|\nabla f(w_t)\|^2 \leq f(w_t) - \mathbb{E}_{s_t}\left[f(w_{t+1})\right] + \alpha_t \beta_{1,t} G_2^2 H_t + \frac{\alpha_t^2 M G_\infty^2 \|\eta_t\|^2}{2} \quad (24)$$

Now, defining $p(t) = \frac{\alpha_t L_t}{Z}$, where $Z = \sum_{t=1}^{T} \alpha_t L_t$, dividing by $Z(1 - \beta_1)$ and summing over $t$:

$$\sum_{t=1}^{T} p(t) \|\nabla f(w_t)\|^2 \leq \frac{1}{Z(1 - \beta_1)} \sum_{t=1}^{T} \left( f(w_t) - \mathbb{E}_{s_t} [f(w_{t+1})] + \alpha_t \beta_{1,t} G_2^2 H_t + \frac{\alpha_t^2 M G_\infty^2 \|\eta_t\|^2}{2} \right)$$

(25)

Now, taking the conditional expectation over all samples $S$ given $Z$:

$$\begin{aligned}
\mathbb{E}_S \left[ \sum_{t=1}^{T} p(t) \|\nabla f(w_t)\|^2 \Big| Z \right] &\leq \frac{1}{Z(1 - \beta_1)} \Big( \sum_{t=1}^{T} \big( \mathbb{E}_S [f(w_t)|Z] - \mathbb{E}_S [\mathbb{E}_{s_t} [f(w_{t+1})] |Z] \big) \\
&\qquad + \sum_{t=1}^{T} \mathbb{E} \left[ \alpha_t \beta_{1,t} G_2^2 H_t + \frac{\alpha_t^2 M G_\infty^2 \|\eta_t\|^2}{2} \Big| Z \right] \Big) \\
&\leq \frac{1}{Z(1 - \beta_1)} \Big( \sum_{t=1}^{T} \big( \mathbb{E}_S [f(w_t)|Z] - \mathbb{E}_S [f(w_{t+1})|Z] \big) \\
&\qquad + \sum_{t=1}^{T} \mathbb{E} \left[ \alpha_t \beta_{1,t} G_2^2 H_t + \frac{\alpha_t^2 M G_\infty^2 \|\eta_t\|^2}{2} \Big| Z \right] \Big) \\
&= \frac{1}{Z(1 - \beta_1)} \Big( f(w_1) - \mathbb{E}_S [f(w_{T+1})|Z] \\
&\qquad + \sum_{t=1}^{T} \mathbb{E} \left[ \alpha_t \beta_{1,t} G_2^2 H_t + \frac{\alpha_t^2 M G_\infty^2 \|\eta_t\|^2}{2} \Big| Z \right] \Big)
\end{aligned}$$

(26)

where in the second step we used $\mathbb{E}_S [\mathbb{E}_{s_t} [f(w_{t+1})] |Z] = \mathbb{E}_S [f(w_{t+1})]$ which follows from the assumption that $p(Z|s_t) = p(Z)$, and the third step follows from a telescoping sum, along with the fact that $\mathbb{E}_S [f(w_1)] = f(w_1)$. Now, using $f(w_1) - \mathbb{E}_S [f(w_{T+1})|Z] \leq f(w_1) - f(w^\star) \leq D$:

$$\mathbb{E}_S \left[ \sum_{t=1}^{T} p(t) \|\nabla f(w_t)\|^2 \Big| Z \right] \leq \frac{1}{Z(1 - \beta_1)} \left( D + \sum_{t=1}^{T} \mathbb{E}_S \left[ \alpha_t \beta_{1,t} G_2^2 H_t + \frac{\alpha_t^2 M G_\infty^2 \|\eta_t\|^2}{2} \Big| Z \right] \right)$$

(27)

Then, taking the expectation over $Z$:

$$\mathbb{E}_S \left[ \sum_{t=1}^{T} p(t) \|\nabla f(w_t)\|^2 \right] \leq \mathbb{E} \left[ \frac{1}{Z(1 - \beta_1)} \sum_{t=1}^{T} \left( \frac{D}{T} + \alpha_t \beta_{1,t} G_2^2 H_t + \frac{\alpha_t^2 M G_\infty^2 \|\eta_t\|^2}{2} \right) \right]$$

(28)

Now, let $\alpha_t = \gamma_t \sqrt{\frac{2D}{TMG_\infty^2}}$:

$$\begin{aligned}
\mathbb{E}_S \left[ \sum_{t=1}^{T} p(t) \|\nabla f(w_t)\|^2 \right] &\leq \mathbb{E}_S \left[ \frac{1}{Z(1 - \beta_1)} \sum_{t=1}^{T} \left( \frac{D}{T} + \gamma_t \beta_{1,t} G_2^2 H_t \sqrt{\frac{2D}{TMG_\infty^2}} + \frac{D}{T} \gamma_t^2 \|\eta_t\|^2 \right) \right] \\
&\leq \mathbb{E} \left[ \frac{D}{T \cdot Z(1 - \beta_1)} \sum_{t=1}^{T} \left( 1 + \gamma_t \beta_{1,t} H_t \sqrt{\frac{2dTG_2^2}{MD}} + \gamma_t^2 \|\eta_t\|^2 \right) \right]
\end{aligned}$$

(29)

where we used the fact that $G_2 \leq G_\infty \sqrt{d}$.

Now, recall that $Z = \sum_{t=1}^{T} \alpha_t L_t = \sqrt{\frac{2D}{TMG_\infty^2}} \sum_{t=1}^{T} \gamma_t L_t$:

$$\mathbb{E}_S \left[ \sum_{t=1}^{T} p(t) \left\| \nabla f(w_t) \right\|^2 \right] \leq \mathbb{E}_S \left[ \frac{1}{(1-\beta_1)} \sqrt{\frac{MDG_\infty^2}{2T}} \cdot \frac{\sum_{t=1}^{T} \left( 1 + \gamma_t \beta_{1,t} H_t \sqrt{\frac{2dTG_2^2}{MD}} + \gamma_t^2 \left\| \eta_t \right\|^2 \right)}{\sum_{t=1} \gamma_t L_t} \right]$$

$$= \frac{1}{(1-\beta_1)} \sqrt{\frac{MDG_\infty^2}{2T}} \cdot \mathbb{E}_S \left[ \frac{\sum_{t=1}^{T} \left( 1 + \gamma_t \beta_{1,t} H_t \sqrt{\frac{2dTG_2^2}{MD}} + \gamma_t^2 \left\| \eta_t \right\|^2 \right)}{\sum_{t=1} \gamma_t L_t} \right]$$

$$(30)$$

Setting $\beta_{1,t} = \beta_1 = 0$ and checking that $\sum_{t=1}^{T} p(t) \left\| \nabla f(w_t) \right\|^2 = \mathbb{E}_{t \sim \mathcal{P}(t|S)} \left[ \left\| \nabla f(w_t) \right\|^2 \right]$:

$$\mathbb{E}_{\substack{S \sim \mathcal{D}^T \\ t \sim \mathcal{P}(t|S)}} \left[ \left\| \nabla f(w_t) \right\|^2 \right] \leq \sqrt{\frac{MDG_\infty^2}{2T}} \cdot \mathbb{E}_{S \sim \mathcal{D}^T} \left[ \frac{\sum_{t=1}^{T} 1 + \gamma_t^2 \left\| \eta_t \right\|^2}{\sum_{t=1} \gamma_t L_t} \right] \qquad (31)$$

Recalling that $L_t = \min_i \eta_{t,i}$ proves the claim.

$\square$

### B.1 THE CASE WITH FIRST-ORDER MOMENTUM

For the case $\beta_{1,t} > 0$, assume that $\beta_{1,t} = \frac{\beta_1}{\sqrt{t}}$ in Equation 30:

$$\mathbb{E}_S \left[ \sum_{t=1}^{T} p(t) \left\| \nabla f(w_t) \right\|^2 \right] \leq \frac{1}{(1-\beta_1)} \sqrt{\frac{MDG_\infty^2}{2T}} \cdot \mathbb{E}_S \left[ \frac{\sum_{t=1}^{T} \left( 1 + \gamma_t H_t \frac{\beta_1}{\sqrt{t}} \sqrt{\frac{2dTG_2^2}{MD}} + \gamma_t^2 \left\| \eta_t \right\|^2 \right)}{\sum_{t=1} \gamma_t L_t} \right]$$

$$\leq \frac{1}{(1-\beta_1)} \sqrt{\frac{MDG_\infty^2}{2T}} \cdot \mathbb{E}_S \left[ \frac{\beta_1 \sqrt{\frac{2dTG_2^2}{MD}} \left( \max_t \gamma_t H_t \right) \sum_{t=1}^{T} \frac{1}{\sqrt{t}} + \sum_{t=1}^{T} \left( 1 + \gamma_t^2 \left\| \eta_t \right\|^2 \right)}{\sum_{t=1} \gamma_t L_t} \right]$$

$$\leq \frac{1}{(1-\beta_1)} \sqrt{\frac{MDG_\infty^2}{2T}} \cdot \mathbb{E}_S \left[ \frac{2T\beta_1 \sqrt{\frac{2dG_2^2}{MD}} \left( \max_t \gamma_t H_t \right) + \sum_{t=1}^{T} \left( 1 + \gamma_t^2 \left\| \eta_t \right\|^2 \right)}{\sum_{t=1} \gamma_t L_t} \right]$$

$$(32)$$

where in the last step we used $\sum_{t=1}^{T} \frac{1}{\sqrt{t}} \leq 2\sqrt{T}$.

Similarly to the guarantee in Equation 31, we can show a $O(1/\sqrt{T})$ convergence rate if we further assume that there exist constants $H$ and $L$ such that $0 < L \leq \eta_{t,i} \leq H < \infty$ for all $i$ and $t$ (i.e., the parameter-wise learning rates are bounded away from zero and also from above), and that $\gamma_t$ is bounded similarly. For example, having $\gamma_t = \gamma$ yields:

$$\mathbb{E}_S \left[ \sum_{t=1}^{T} p(t) \left\| \nabla f(w_t) \right\|^2 \right] \leq \frac{1}{(1-\beta_1)} \sqrt{\frac{MDG_\infty^2}{2T}} \cdot \mathbb{E}_S \left[ \frac{2TH\gamma\beta_1 \sqrt{\frac{2dG_2^2}{MD}} + \sum_{t=1}^{T} \left( 1 + \gamma^2 dH^2 \right)}{\sum_{t=1} \gamma L} \right]$$

$$= \frac{1}{L(1-\beta_1)} \sqrt{\frac{MDG_\infty^2}{2T}} \cdot \left( \gamma^{-1} + 2H\beta_1 \sqrt{\frac{2dG_2^2}{MD}} + \gamma dH^2 \right)$$

$$(33)$$

where in the first step we used $\left\| \eta_t \right\|^2 \leq dH^2$, $H_t \leq H$, and $L_t \geq L$.

Note that for any constant $\gamma$ the above is $O(1/\sqrt{T})$.

### B.2 THE CASE WITH UNCONDITIONAL DISTRIBUTION OVER ITERATES

To show a similar bound without the assumption that $p(Z|s_t) = p(Z)$, we can alternatively bound $L_t$ and $\|\eta_t\|$ using the worst case over possible samples $S$. From (24) we have, with $\beta_{1,t} = 0$:

$$\left(\inf_S L_t\right) \alpha_t (1 - \beta_1) \|\nabla f(w_t)\|^2 \leq f(w_t) - \mathbb{E}_{s_t}\left[f(w_{t+1})\right] + \frac{\alpha_t^2 LG_\infty^2 \left(\sup_S \|\eta_t\|^2\right)}{2} \tag{34}$$

Now, define $p(t) = \alpha_t \left(\inf_S L_t\right)/Z$ with $Z = \sum_{t=1}^T \alpha_t \left(\inf_S L_t\right)$ instead. As long as $\alpha_t$ does not depend on $S$, $Z$ is no longer a random variable. Following the same steps as above leads to the following:

$$\mathbb{E}_{\substack{S \sim \mathcal{D}^T \\ t \sim \mathcal{P}(t)}}\left[\|\nabla f(w_t)\|^2\right] \leq \sqrt{\frac{MDG_\infty^2}{2T} \cdot \frac{\sum_{t=1}^T 1 + \gamma_t^2 \left(\sup_S \|\eta_t\|^2\right)}{\sum_{t=1}^T \gamma_t \left(\inf_S L_t\right)}} \tag{35}$$

In particular, if there are constants $H$ and $L$ such that $0 < L \leq \eta_{t,i} \leq H < \infty$ for all $i$ and $t$, can bound $\sup_S \|\eta_t\|^2 \leq dH^2$ and $\inf_S L_t \geq L$, yielding:

$$\mathbb{E}_{\substack{S \sim \mathcal{D}^T \\ t \sim \mathcal{P}(t)}}\left[\|\nabla f(w_t)\|^2\right] \leq \sqrt{\frac{MDG_\infty^2}{2T} \cdot \frac{\sum_{t=1}^T 1 + \gamma_t^2 dH^2}{L \sum_{t=1}^T \gamma_t}} \tag{36}$$

Hence a $O(1/\sqrt{T})$ follows as long as $\gamma_t$ can be upper and lower bounded accordingly by constants.

## C   EXPERIMENTAL DETAILS

### C.1   CIFAR

The CIFAR-10 and CIFAR-100 datasets (Krizhevsky, 2009) consist of 60,000 RGB images with $32 \times 32$ pixels, with a standard split of 50,000 and 10,000 training and test images, sampled from 10 and 100 classes, respectively. We pre-process the dataset with channel-wise normalization using statistics from the training set, and for data-augmentation we flip each image horizontally with $50\%$ probability, along with a random cropping, achieved by first padding 4 black pixels to each edge of the image and then extracting a random $32 \times 32$ crop.

For the hyperparameter exploration with a Wide ResNet 28-4, we measure performance on a validation set of 5,000 images sampled from the training set, while for the final results with the Wide ResNet 28-10, we report results on the test set.

Each Wide ResNet is trained with a batch size of 128 for a total of 200 epochs on a single GPU, with the learning rate being decayed by a factor of 5 at epochs 60, 120 and 160. A weight decay of 0.0005 is used with every optimization method. For SGD, we use a momentum of 0.9, while for adaptive methods, we use $\beta_1 = 0.9, \beta_2 = 0.999$, as suggested in their respective papers. For AdaBound, we use the default $\alpha^* = 0.1$ and $\gamma = 10^{-3}$. For AdaShift, we use the default $n = 10$.

### C.2   IMAGENET

The ILSVRC 2012 datasets (Russakovsky et al., 2015) is composed of 1.2M training and 50,000 validation RGB images, sampled from a total of 1,000 classes. We train a ResNet 50 (He et al., 2016b), and report results using single-crops of $224 \times 224$ images for evaluation. We follow Gross & Wilber (2016) for data-augmentation and training specifics: in particular, we decay the learning rate by a factor of 10 at epochs 30, 60 and 90, and use a weight decay of 0.0001. We use 4 GPUs for training, with a total batch size of 256 (64 per GPU).

### C.3   PENN TREEBANK

We follow Merity et al. (2018) for both the model and optimization details for the Penn Treebank dataset (Marcus et al., 1994) for character-level language modelling (Mikolov et al., 2010). Again,

we measure validation performance for the small-scale experiments, and test performance for the large-scale ones.

More specifically, we train a 3-layer LSTM (Hochreiter & Schmidhuber, 1997) with an embedding size of 200, and decay the learning rate by a factor of 10 at epochs 300 and 400, while the model is trained for a total of 500 epochs. We use a batch size of 128 and BPTT length of 150. The model is regularized with a weight decay of $1.2 \times 10^{-6}$, a weight dropout (Merity et al., 2017) of $p = 0.5$, and a dropout (Srivastava et al., 2014) of $p = 0.1$ for the input and output layers and $p = 0.25$ for the hidden-layer.

