# OpenReview forum: "Domain-Independent Dominance of Adaptive Methods"
_ICLR.cc/2020/Conference — Reject_

### Official Review · AnonReviewer2 · 2019-10-22
**Official Blind Review #2**

**Rating:** 3

**Review:**

Summary:
This paper proposes a new adaptive method, which is called AvaGrad. The authors first show that Adam may not converge to a stationary point for a stochastic convex optimization in Theorem1, which is closely related to [1]. They then show that by simply making $eta_t$ to be independent of the sample $s_t$, Adam is able to converge just like SGD in Theorem2. Theorem2 follows the standard SGD techniques. Next, they propose AVAGRAD, which is based on the idea of getting rid of the effect of $\epsilon$.

Strength:
The experiment results are impressive. They show that Adam can outperform SGD on vision tasks.
Recently, people have found out that $\epsilon$ is a very sensitive hyper-parameter. It is good to see some research directly addresses this problem.

Weakness:
The word "domain" is confusing.
If the Adam-type algorithms are the delayed version in Table 1?
It is not compatible with AdamW.
The results on the image datasets seem too good to be true.

Implementation Issue:
***Many implementation details in the below discussion are different from the paper (e.g. hyperparameters and network architecture). So the following experiment results may not be used for assessment of the quality of the proposed method.***
I tried the proposed Delayed Adam on CIFAR-10 using the codebase in (https://github.com/LiyuanLucasLiu/RAdam/tree/master/cifar_imagenet). The performance seems the same as Adam. Delayed Adam even leads to a *divergence* problem, especially with a large learning rate (0.03). The divergence problem never happens when using Adam and AdamW with the same hyperparameters.
Implementation details:
1. Replace the optimizer with the original PyTorch Adam implementation. (https://github.com/pytorch/pytorch/blob/master/torch/optim/adam.py)
2. Swap line 96 and 108 as suggested in the paper.
3. Modified line 89 (bias_correction2=1 - beta2 ** (state['step']-1))
4. Do not run line 97-107 when state['step']==1.
5. Run the following code: python cifar.py -a resnet --depth 20 --epochs 164 --schedule 81 122 --gamma 0.1 --wd 1e-4 --optimizer adam  --beta1 0.9 --beta2 0.999  --checkpoint ./logdir --gpu-id 0 --model_name adam_003 --lr 0.03

If the authors can provide more implementation details, I would promote my rating.
e.g.,
1. Are you still using bias correction in the proposed method? If so how do you use them?
2. Do you update the model for the first step?

Reference:
[1] On the convergence of adam and beyond


**Experience Assessment:**

I have published one or two papers in this area.

**Review Assessment: Checking Correctness Of Derivations And Theory:**

I assessed the sensibility of the derivations and theory.

**Review Assessment: Checking Correctness Of Experiments:**

I assessed the sensibility of the experiments.

**Review Assessment: Thoroughness In Paper Reading:**

I read the paper at least twice and used my best judgement in assessing the paper.

---

> ### Author Response · Authors · 2019-11-13
> **Response to reviewer 2 [2/2]**
>
> “Implementation Issue”
>
> The learning rates of Adam and Delayed Adam are not equivalent. As an informal motivation to see this, consider a sequence of many samples with small gradients, followed by a sample with a large gradient. For Adam, the large gradient will be immediately used to update v_t, which will increase, hence decreasing \eta_t and consequently also the step size. On the other hand, for Delayed Adam, the large gradient will not affect \eta_t, but only \eta_{t+1}, hence the step size will be larger than the one which Adam would have computed. In practice, to achieve similar behavior with Delayed Adam, one should use a smaller learning rate than the one used with Adam.
>
> We have performed runs using the same codebase (https://github.com/LiyuanLucasLiu/RAdam/tree/master/cifar_imagenet) so that results can be more easily replicated. For Delayed Adam, we performed the steps 2 to 4 in your review so that we have a matching implementation (which is equivalent to the one used in the experimental results in our paper), referred to as ‘dadam’ when chosen by command-line arguments below. To control the epsilon parameter, we added a command-line argument args.eps:
>
> parser.add_argument('--eps', default=1e-8, type=float, help='epsilon parameter for adaptive methods')
>
> Which is used to instantiate the adaptive methods, e.g.:
>
> optimizer = optim.Adam(model.parameters(), lr=args.lr, betas=(args.beta1, args.beta2), weight_decay=args.weight_decay, eps=args.eps)
>
> To facilitate reproducibility, we remove cudnn.benchmark=True in cifar.py (right after model = torch.nn.DataParallel(model).cuda()), replacing it with:
>
> cudnn.deterministic = True
> cudnn.benchmark = False
>
> We also use a manual seed 0 for all experiments. The commands, followed by the given results, were:
>
> Adam, lr=0.001, eps=1e-8
> python cifar.py -a resnet --depth 20 --epochs 164 --schedule 81 122 --gamma 0.1 --wd 1e-4 --optimizer adam  --beta1 0.9 --beta2 0.999  --checkpoint ./logdir --gpu-id 0 --model_name adam_001 --lr 0.001 --manualSeed 0 --eps 1e-8
> Best acc: 90.86%
>
> Delayed Adam, lr=0.001, eps=1e-8
> python cifar.py -a resnet --depth 20 --epochs 164 --schedule 81 122 --gamma 0.1 --wd 1e-4 --optimizer dadam  --beta1 0.9 --beta2 0.999  --checkpoint ./logdir --gpu-id 0 --model_name dadam_001 --lr 0.001 --manualSeed 0 --eps 1e-8
> Best acc: 90.69%
>
> These show that Adam and Delayed Adam perform similarly given standard values for the learning rate.
>
> Regarding the ‘too good to be true results’, here there are commands with hyperparameter settings showing Adam can outperform SGD for ResNet training on CIFAR. We used the same values for the learning rate and epsilon as in our paper — no extra hyperparameter search was performed.
>
> Adam, lr=0.1, eps=0.1
> python cifar.py -a resnet --depth 20 --epochs 164 --schedule 81 122 --gamma 0.1 --wd 1e-4 --optimizer adam  --beta1 0.9 --beta2 0.999  --checkpoint ./logdir --gpu-id 0 --model_name adam_01 --lr 0.1 --manualSeed 0 --eps 0.1
> Best acc: 92.21%
>
> SGD, lr=0.1
> python cifar.py -a resnet --depth 20 --epochs 164 --schedule 81 122 --gamma 0.1 --wd 1e-4 --optimizer sgd  --checkpoint ./logdir --gpu-id 0 --model_name sgd_01 --lr 0.1 --manualSeed 0
> Best acc: 91.91%
>
> To further facilitate reproducibility, we are preparing a codebase to be publicly available with our implementation of AvaGrad and the code to run our experiments.

---

> ### Author Response · Authors · 2019-11-13
> **Response to reviewer 2 [1/2]**
>
> Thank you for the review. We address your points individually below — please let us know if we can clarify or address any further concerns.
>
>
>
> "If the Adam-type algorithms are the delayed version in Table 1?”
>
> No, the adaptive methods in Table 1 are not the delayed versions. In Table 1, the only method that applies a delay in the computation of v_t is AvaGrad (and, consequently, AvaGradW). Delayed Adam was only used in the synthetic experiment of Figure 1, for the purpose of theoretical motivation and evaluation.
>
> ———————————————-
>
> “It is not compatible with AdamW”
>
> There is no issue of compatibility: we are proposing a new optimizer, not modifying existing ones. We do not modify Adam or AdamW in any form for the results in Table 1: other than AvaGrad and AvaGradW, we use PyTorch built-in or official implementations for each method, and the results were achieved when running with hyperparameters found after extensive grid search (Figures 2 and 3).
>
> The optimizer referred as ‘AvaGradW’ in Table 1, which yielded the best results on Penn Treebank, is the result of applying weight decay as in Loshchilov & Hutter to AvaGrad, our newly proposed optimizer.
>
> ———————————————-
>
> “Are you still using bias correction in the proposed method?”
>
> We use bias correction similarly to Adam, dividing m_t by 1 - \beta_1^t, and v_{t-1} by 1 - \beta_2^{t-1}.The norm of v_{t-1}, used to scale the learning rate, is computed from the bias-corrected v_{t-1}.
>
> ———————————————-
>
> “Do you update the model for the first step?”
> We do not update the model in the first step.
>
> ———————————————-
>
> “The results on the image datasets seem too good to be true.”
>
> Our results are correct as reported and highlight the value of our proposed optimizer, which improves results across different datasets and architectures.
>
> To achieve improved results with existing optimizers, we had to search over a hyperparameter space larger than typically employed, which may be why those best case results appear surprisingly good. See an example run with hyperparameter settings where Adam outperforms SGD at the end of our reply.

---

### Official Review · AnonReviewer3 · 2019-10-23
**Official Blind Review #3**

**Rating:** 6

**Review:**

In this paper the authors develop variants of Adam which corrects for the relationship of the gradient and adaptive terms that causes convergence issues, naming them Delayed Adam and AvaGrad. They also provide proofs demonstrating they solve the convergence issues of Adam in O(1/sqrt(T)) time. They also introduce a convex problem where Adam fails to converge to a stationary point.

This paper is clearly written and has reasonable experimental support of its claims. However in terms of novelty, AdaShift was published at ICLR last year (https://openreview.net/forum?id=HkgTkhRcKQ) and seems to include a closely related analysis and update rule of your proposed optimizers. In AdaShift instead of correcting for the correlation between the gradient and eta_t, they correct for the relationship between the gradient and the second moment term v_t. Could you further clarify the differences between the two, both in your approach to deriving the new update rule and the algorithms themselves? Additionally, their Theorem 1 could be compared to yours, but noting the differences for these seems less important. If the optimizers are unique enough, including AdaShift in your experiments would be very useful for demonstrating their differences.

Regarding experiments, while it is true that adaptive methods are supposed to be less “sensitive” to hyperparameter choices, the limits of the feasible ranges for each hyperparameter could vary drastically across problems (especially, as previously demonstrated, across different batch sizes.) Thus, not retuning across experiments seems like it could negatively affect performance for any of the transferred hyperparameter settings. Instead of demonstrating hyperparameter insensitivity by carrying over hyperparameter settings, one could instead retune for each problem and show that a higher percent of hyperparameter combinations result in the same/similar best performance (similar to what is done in Fig. 2, but also showing a (1-dimensional) SGD version which would presumably contain fewer high performing settings.)

Some additional comments:
-The contribution of Theorem 1 is a nice addition to the literature.
-Your tuning of epsilon is great! I believe more papers should include epsilon in their hyperparameter sweeps.
-Scaling epsilon with step size makes sense when considering that Adam is similar to a trust region method, where epsilon is inversely proportional to the trust region radius. However, in section 5 implying that epsilon should be as large as possible in the worst case seems like an odd result given that this would always diminish your second moment term as much as possible, defeating the point of the additional Adam-like adaptivity. Can you comment on why this diminished adaptivity would be desirable in the worst case scenario analyzed?
-The synthetic toy problem is much appreciated, more papers should start with a small, interpretable experiment.
-Was SGD with momentum used? If not, this may not be a fair comparison, as I believe it is much more common to use momentum with SGD. If momentum was used, was the momentum hyperparameter tuned? If not, this may be advantageous to the Adam based methods, as they have more versatile adaptability and thus may not need as much care with their selection of momentum values.
-Was a validation set used for CIFAR? You note in appendix C that there are 50k train and 10k test. You mention validation performance in the main text, so this is just double checking.
-The demonstration in figures 2, 3 of decoupling the step size and epsilon in interesting! Given that the best performing values seem to be on the edges of the ranges tested, I would be curious if the trend continues for more extreme values of alpha and epsilon (one could sparsely search along the predicted trendlines.)

Nits:
-“Vanilla SGD is still prevalent, in spite of the development of seemingly more sophisticated adaptive alternatives...” This could use some citations to back up the claim, because as far as I know it is much more common to use SGD with momentum and is actually rare to use vanilla SGD (the DenseNets and Resnets citations use momentum=0.9.)
-It would be nice to highlight in color the diff from vanilla Adam in the Algorithm sections.
-It is not super clear from the text how in eq 26 you get \sum{E[f(w_t)|Z]} = f(w_1)
-I may be misreading something, but I believe the H in the leftmost term in the last line of eq 33 should be an L.
-In section 5, “for a fixed learning rate (as is typically done in practice, except for discrete decays during training)” seems like an overly broad claim, given that authors commonly use polynomial, linear, exponential, cosine, or other learning rate decay/warmups. Granted for some CIFAR and ImageNet benchmarks there are more common discrete learning rate schedules, but that does not seem to be the overwhelmingly prevalent technique.

Overall, while this area of analyzing Adam and proposing modifications is a popular and crowded subject, I believe this paper may contribute to it if my concerns are addressed. While I currently do not recommend acceptance, I am open to changing my score after considering the author comments!


**Experience Assessment:**

I have published one or two papers in this area.

**Review Assessment: Checking Correctness Of Derivations And Theory:**

I carefully checked the derivations and theory.

**Review Assessment: Checking Correctness Of Experiments:**

I carefully checked the experiments.

**Review Assessment: Thoroughness In Paper Reading:**

I read the paper thoroughly.

---

> ### Author Response · Authors · 2019-11-13
> **Response to reviewer 3 [2/2]**
>
> “Can you comment on why this diminished adaptivity would be desirable in the worst case scenario analyzed?”
>
> Exactly characterizing why adaptivity is undesirable in theory is beyond the scope of this paper, but the same observation is present in previous papers in the literature, for example:
>
> Staib et al. - Escaping Saddle Points with Adaptive Gradient Methods (check Section 5.1)
> De et al. - Convergence guarantees for RMSProp and ADAM in non-convex optimization and an empirical comparison to Nesterov acceleration
>
> ———————————————-
>
> “Was SGD with momentum used?”
>
> We used SGD with a Nesterov momentum of 0.9, following seminal works such as the ResNet (He et al.’15), DenseNet (Huang et al.’16), Wide ResNet (Zagoruyko & Komodakis’16) and ResNeXt (Xie et al.’16) papers.
>
> ———————————————-
>
> “Was a validation set used for CIFAR?”
>
> We used a validation set of 5k examples sampled from the training set for the hyperparameter search and the results in Figure 2 and 3. For the final results in Table 1, we used the actual test set composed of 10k images. We have updated the paper to clarify this.
>
> ———————————————-
>
> “I would be curious if the trend continues for more extreme values of alpha and epsilon”:
>
> We have performed preliminary experiments and observed that the trend indeed continues for extreme values of alpha and epsilon when training with Adam or AMSGrad. In particular, the performance with alpha = c * epsilon for some fixed c is nearly identical regardless of alpha and epsilon, as long as they are large enough. For a motivation why this happens, note that if epsilon is large, then \eta_t = 1 / (\sqrt(v_t) + \epsilon) is approximately 1 / \epsilon. If we set \alpha = c * \epsilon, then the update rule becomes w_{t+1} = w_t - c * \epsilon * g_t / \epsilon = w_t - c * g_t, regardless of \alpha and \epsilon.
>
> ———————————————-
>
> “it is much more common to use SGD with momentum”
>
> By ‘vanilla SGD’ we meant SGD with momentum. We use SGD with momentum for all experiments in the paper. We have changed ‘vanilla SGD’ to ‘SGD’ in the paper.
>
> ———————————————-
>
> “It would be nice to highlight in color the diff from vanilla Adam in the Algorithm sections.”
>
> We have put the differences in red in the revised version of the paper.
>
> ———————————————-
>
> “It is not super clear from the text how in eq 26 you get \sum{E[f(w_t)|Z]} = f(w_1)”
>
> Here, f(w_1) - E[f(w_{T+1})|Z] is the result of a telescoping sum (more specifically, the sum in the right-hand side of the first line in eq 26). First, we have that E_S[E_{s_t}[f(w_{t+1})]|Z] = E_S[f(w_{t+1})] due to the assumption. With this, the sum has the form \sum_{t=1}^T a_t - a_{t+1} (where a_t is the expectation of f(w_t)) and the sum of all terms equals to a_1 - a_{T+1} by telescoping sum. Since w_1 does not depend on how points are sampled, we have E[f(w_1)] = f(w_1), finally yielding f(w_1) - E[f(w_{T+1})|Z]. We have updated the paper to make this clearer, adding an additional step in the derivation.
>
> ———————————————-
>
> “I believe the H in the leftmost term in the last line of eq 33 should be an L.”
>
> This was indeed a typo, which we have fixed in the revised version — thanks for pointing it out.
>
> ———————————————-
>
> “authors commonly use polynomial, linear, exponential, cosine, or other learning rate decay/warmups”
>
> We have changed ‘typically’ to ‘not uncommonly’ to be more precise. In this statement we were referring specifically to the papers that presented our baselines (Zagoruyko & Komodakis, Merity et al.), which all use a step-wise learning rate schedule.

---

> > ### Comment · AnonReviewer3 · 2019-11-14
> > **Response to response to reviewer 3 [*/2]**
> >
> > Thank you very much for the detailed responses! These are very thorough in addressing the concerns raised, I appreciate (and sympathize with the work involved in) the quick turn around in adding additional experiments. I will definitely be raising my score.

---

> ### Author Response · Authors · 2019-11-13
> **Response to reviewer 3 [1/2]**
>
> Thank you for the review and your comments. We address your points individually below — please let us know if we can clarify or address any further concerns.
>
>
>
> “AdaShift was published at ICLR last year and seems to include a closely related analysis and update rule”
>
> The differences in the analysis and update rule are significant. AvaGrad, our main contribution, includes the convergence fix of Delayed Adam — a delay in the computation of v_t —  but, more significantly, also applies a new adaptive scaling factor to gradients.  This new adaptive scaling rule is responsible for AvaGrad’s superior performance on real datasets and its better hyperparameter separation.
>
> Contrasting Delayed Adam with AdaShift, the update rule in AdaShift applies a delay in the computation of v_t and, at the same time, a limited horizon on the computation of m_t. If we set n=1 and \phi as the identity function, then we recover an update rule that is similar to Delayed Adam, but with \beta_1 = 0 (that is, we lose first-order momentum). In the general setting where n>1, AdaShift requires storing a history of the past n gradients, and there is little relation to the update rule of Delayed Adam.
>
> AvaGrad, and not Delayed Adam, is our actual contribution in terms of a new adaptive method. AvaGrad’s key significance is that the parameter-wise learning rates \eta_t are normalized, which is precisely what decouples the learning rate and epsilon. In our presentation, Delayed Adam serves as a motivation for the design of AvaGrad: the normalization of \eta is inspired by the convergence rate of Delayed Adam.
>
> In terms of analysis, Zhou et al. analyze AdaShift in the online convex optimization framework, while we provide convergence rates in the smooth stochastic non-convex setting: both the implications and the proof technique differ significantly between the two.
>
> ———————————————-
>
> “Could you further clarify the differences between the two”
>
> For any update rule where the gradient and \eta are uncorrelated, we can use Theorem 2 (considering standard assumptions) to assure convergence regardless of how \eta is computed. On the other hand, if \eta has a different form, then having the gradient and v_t to be uncorrelated might not be enough to guarantee convergence. In other words, to guarantee convergence, correcting for the gradient and \eta is sufficient for any functional form of \eta, while correcting for the gradient and v_t is might not be sufficient (it is sufficient, however, when \eta_t is only a function of v_t).
>
> ———————————————-
>
> “their Theorem 1 could be compared to yours”
>
> Although this might be a minor concern, the differences are significant and important: their Theorem 1 is a statement about regret in the online convex optimization framework, while our Theorem 1 is about stationarity in the stochastic smooth non-convex setting.
>
> ———————————————-
>
> “including AdaShift in your experiments would be very useful for demonstrating their differences”
>
> Thanks for the suggestion -- we have now added AdaShift to our experiments. Following the same protocol we used for all adaptive methods, we first performed grid search over the learning rate and epsilon on CIFAR with a Wide ResNet 28-4, where AdaShift performed best with lr = epsilon = 1.0. Next, we trained a Wide ResNet 28-10, where AdaShift achieved 4.08% and 18.88% error on CIFAR10 and CIFAR100 (outperforming every adaptive method except for AvaGrad on the latter). Following the same protocol n Penn Treebank, it yielded a bpc of 1.274 with a 4-layer LSTM with 1000 units per layer. We won’t have ImageNet results for AdaShift by the rebuttal deadline, but we will add them to the camera-ready version of the paper.
>
> ———————————————-
>
> “retuning across experiments”
>
> This is an important observation. We re-tuned the learning rate of SGD for CIFAR 10/100  with the Wide ResNet 28-10, and for PTB with the 4x1000 LSTM, yielding the following performance for each learning rate:
>
> C10 (val error):
> 1.0:  	8.84%
> 0.5:  	4.98%
> 0.1:  	3.86%
> 0.05: 	4.20%
> 0.01: 	5.14%
>
> C100 (val error):
> 1.0:  	37.13%
> 0.5:  	23.25%
> 0.1:  	19.05%
> 0.05: 	19.92%
> 0.01: 	22.51%
>
> PTB (bits per character, lower is better):
> 100.0: 	1.473 bpc
> 20.0:  	1.238 bpc
> 10.0:  	1.253 bpc
> 2.0:		1.298 bpc
> 1.0: 		1.348 bpc
>
> The best learning rates, even when re-tuning SGD on the large experiments, were the same as the ones we used to get the results in Table 1. Performing the same level of tuning we did for the Wide ResNet 28-4 and 4x300 LSTM models for adaptive methods is unfeasible, as it consisted of over 150 runs for each algorithm. Since adaptive methods, even when not re-tuned, still outperform a re-tuned SGD, our findings remain consistent.

---

### Official Review · AnonReviewer4 · 2019-10-24
**Official Blind Review #4**

**Rating:** 3

**Review:**

In this paper, the authors present a new adaptive gradient method AvaGrad. The authors claim the proposed method is less sensitive to its hyperparameters, compared to previous algorithms, and this is due to decoupling the learning rate and the damping parameter.

Overall, the paper is well written, and is on an important topic. However, I have a few concerns about the paper, which I will list below.

1. The fact that adaptive gradient methods converge with a fast rate when the sum in the denominator is taken till the t-1th iterate has appeared in previous papers before [1]. The convergence rate analysis for this case is fairly simple, and I am not sure if analyzing RMSProp/Adam in this setting should be considered a significant contributions of the paper.

2. The proposed algorithm AvaGrad is a simple but interesting idea. I have a number questions about the experimental evaluation though, which makes it hard for me to evaluate the significance of the results presented:

a) Was momentum used with SGD?

b) How is the optimal hyperparameters (learning rate and damping, i..e, epsilon, parameters) selected?

c) Do any of these conclusions change when trying out a very small or very large batch size?

d) I am not convinced that using the same optimal hyperparams as the WRN-28-4 task on the WRN-28-10 and ResNet50 models is a reasonable experiment. Why is this a good idea? While this does support the claim that adaptive gradient methods are less sensitive to hyperparameter settings, but makes the other claim about AvaGrad generalizing just as well as SGD weaker?

e) One of the key claims that adaptive gradient methods generalize better when using a large damping (epsilon) parameter has appeared in previous papers as well [2, 3].


Overall, in my view, this is a borderline paper mostly because I think a number of the results presented have been shown in other recent papers. My score reflects this. However, I think decoupling the learning rate and the damping parameter by normalizing the preconditioner is a simple but interesting idea, and I am willing to increase my score based on the discussion with the authors and other reviewers.


[1] X. Li and F. Orabona. On the Convergence of Stochastic Gradient Descent with Adaptive Stepsizes. In AISTATS 2019
[2] M. Zaheer, S. Reddi, D. Sachan, S. Kale, and S. Kumar. Adaptive methods for nonconvex optimization. in NeurIPS 2018.
[3] S. De, A. Mukherjee, and E. Ullah. Convergence guarantees for rmsprop and adam in non-convex optimization and an empirical comparison to nesterov acceleration. arXiv:1807.06766, 2018.

A few more minor comments:

1. The authors say that methods like AMSGrad fail to match the convergence rate of SGD. But this statement seems misleading since it is not clear whether the worse rate is due to the analysis (which gives an upper bound) or the algorithm?

2. In the related work section, the authors discuss convergence rates of algorithms with constant and decreasing step sizes together. This can be confusing to the reader, and it is best to explicitly mention the setting under which the different results were derived.

=======================================

Edit after rebuttal:
I thank the authors for the detailed response and for the updated version of the paper. After discussion with other reviewers, we are still not convinced that the hyperparameter tuning in the experiments (especially the baselines) is rigorous enough. This is especially important given that this paper proposes a new optimizer. We are also concerned about the novelty of the results, and believe most of these results have appeared in previous work. So I am not increasing my score. I would encourage the authors to do a more rigorous experimental evaluation of the proposed algorithm.

**Experience Assessment:**

I have published in this field for several years.

**Review Assessment: Checking Correctness Of Derivations And Theory:**

I assessed the sensibility of the derivations and theory.

**Review Assessment: Checking Correctness Of Experiments:**

I carefully checked the experiments.

**Review Assessment: Thoroughness In Paper Reading:**

I read the paper at least twice and used my best judgement in assessing the paper.

---

> ### Author Response · Authors · 2019-11-13
> **Response to reviewer 4 [2/2]**
>
> “using the same optimal hyperparams as the WRN-28-4 task (...) makes the other claim about AvaGrad generalizing just as well as SGD weaker”
>
> This is an important observation. We re-tuned the learning rate of SGD for CIFAR 10/100  with the Wide ResNet 28-10, and for PTB with the 4x1000 LSTM, yielding the following performance for each learning rate:
>
> C10 (val error):
> 1.0:  	8.84%
> 0.5:  	4.98%
> 0.1:  	3.86%
> 0.05: 	4.20%
> 0.01: 	5.14%
>
> C100 (val error):
> 1.0:  	37.13%
> 0.5:  	23.25%
> 0.1:  	19.05%
> 0.05: 	19.92%
> 0.01: 	22.51%
>
> PTB (bits per character, lower is better):
> 100.0: 	1.473 bpc
> 20.0:  	1.238 bpc
> 10.0:  	1.253 bpc
> 2.0:		1.298 bpc
> 1.0: 		1.348 bpc
>
> The best learning rates, even when re-tuning SGD on the large experiments, were the same as the ones we used to get the results in Table 1. Performing the same level of tuning we did for the Wide ResNet 28-4 and 4x300 LSTM models for adaptive methods is unfeasible, as it consisted of over 150 runs for each algorithm. Since adaptive methods, even when not re-tuned, still outperform a re-tuned SGD, our findings remain consistent.
>
> In practice, it is common to tune hyperparameters in smaller models and translate these to large-scale experiments — moreover, the hyperparameters used for SGD coincide with the ones extensively used in the literature, e.g. the ResNet (He et al.’15), DenseNet (Huang et al.’16), Wide ResNet (Zagoruyko & Komodakis’16) and ResNeXt (Xie et al.’16) papers.
>
> ———————————————-
>
> “One of the key claims that adaptive gradient methods generalize better when using a large damping (epsilon) parameter has appeared in previous papers as well [2, 3].”
>
> The values for epsilon explored in [2] and [3] are at most 1e-3, a value 100 times smaller than the one we found to be optimal for Adam/AMSGrad and 10,000 times smaller than the optimal one for AvaGrad/AdamW.
>
> To clarify our contribution: we observe that the optimal hyperparameters for popular adaptive methods can be many orders of magnitude larger than the ones typically explored in the literature. Tuning them to such extreme ranges had not been done before due to the computational costs of grid search, especially since the optimal values for epsilon are strongly coupled with the learning rate (Figure 2 and 3, left plots) in a non-linear manner (more specifically, there are two visible ‘regimes’ that determine how epsilon and the learning rate interact).
>
> Our proposed method, AvaGrad, effectively decouples the two (Figure 2 and 3, right plots), hence hyperparameter tuning can be broken into two line searches. The precise interaction between the learning rate and epsilon, as shown in our figures, has not appeared in previous works, nor has an effective method to tune both hyperparameters without yielding quadratic time complexity. Lastly, showing that adaptive methods — even Adam —  can outperform SGD on ImageNet, without extra caveats such as involved warmup schedules, is an important and novel experimental result.
>
> ———————————————-
>
> “it is not clear whether the worse rate is due to the analysis” / “can be confusing to the reader, and it is best to explicitly mention the setting under which the different results were derived”
>
> We agree that these can be sources of confusion to the readers, and we have clarified this in the revised version of the paper.

---

> ### Author Response · Authors · 2019-11-13
> **Response to reviewer 4 [1/2]**
>
> Thank you for the review and your comments. We address your points individually below — please let us know if we can clarify or address any further concerns.
>
>
>
> “I am not sure if analyzing RMSProp/Adam in this setting should be considered a significant contributions of the paper”
>
> The convergence rate of Delayed Adam is not the main point of the paper. Its form — which depends explicitly on the norm of \eta — is what motivates the design of AvaGrad, hence the role of the analysis was to, first of all, inspire the design of a better adaptive method. AvaGrad, a new adaptive algorithm with different characteristics and increased performance, is the major contribution of the paper.
>
> ———————————————-
>
> “Was momentum used with SGD?”
>
> We used SGD with nesterov momentum of 0.9, closely following Zagoruyko & Komodakis for the CIFAR Wide ResNet experiments,  Merity et al. for the Penn Treebank LSTM experiments, and He et al. for the ResNet ImageNet experiments.
>
> ———————————————-
>
> “How is the optimal hyperparameters (learning rate and damping, i..e, epsilon, parameters) selected?”
>
> We first performed extensive grid search on smaller-scale experiments, training a Wide ResNet 28-4 on CIFAR, and a 4-layer, 300 units per layer LSTM on Penn Treebank, evaluating the performance on the validation set — the results for Adam and AvaGrad are given in Figure 2 and Figure 3. Next, we used the values that yielded the best validation performance to train a Wide ResNet 28-10 on CIFAR, a 4-layer, 1000 units LSTM on Penn Treebank, and a ResNet 50 on ImageNet, achieving the results presented in Table 1. The protocol is described in Section 6.2 (paragraph 3 and the two following bullet points) and Section 6.3 (paragraph 2).
>
> In particular, for CIFAR we searched over:
>
>   - For AvaGrad and AvaGradW:
> Learning rate in {0.0005, 0.001, 0.005, 0.01, 0.05, 0.1, 0.5, 1.0, 5.0, 10.0, 100.0, 500.0, 1000.0, 5000.0}
> Epsilon in {1e-8, 1e-7, 1e-6, 1e-5, 1e-4, 1e-3, 1e-2, 1e-1, 1.0, 10.0, 100.0}
>
>   - For Adam, AMSGrad, AdamW, AdaBound:
> Learning rate in {0.00005, 0.00001, 0.0005, 0.001, 0.005, 0.01, 0.05, 0.1, 0.5, 1.0, 5.0, 10.0, 100.0, 500.0}
> Epsilon in {1e-8, 1e-7, 1e-6, 1e-5, 1e-4, 1e-3, 1e-2, 1e-1, 1.0, 10.0, 100.0}
>
> The best learning rate and epsilon, for each method, was:
>
> Adam (0.1, 0.1)
> AMSGrad (0.1, 0.1)
> AvaGrad (1.0, 10.0)
> AvaGradW (1.0, 0.1)
> AdamW(0.5, 10.0)
> AdaBound(0.005, 0.01)
>
> For Penn Treebank, we searched over:
>
>   - For AvaGrad and AvaGradW:
> Learning rate in {0.2, 2.0, 20.0, 200.0, 2000.0}
> Epsilon in {1e-8, 1e-7, 1e-6, 1e-5, 1e-4, 1e-3, 1e-2, 1e-1, 1.0, 10.0, 100.0}
>
>   - For Adam, AMSGrad, AdamW, AdaBound:
> Learning rate in {0.0002, 0.002, 0.02, 0.2, 2.0, 20.0}
> Epsilon in {1e-8, 1e-7, 1e-6, 1e-5, 1e-4, 1e-3, 1e-2, 1e-1, 1.0, 10.0, 100.0}
>
> The best learning rate and epsilon (\alpha, \epsilon), for each method, was:
>
> Adam (0.002, 1e-8)
> AMSGrad (0.002, 1e-8)
> AvaGrad (200, 1e-8)
> AvaGradW (200, 1e-6)
> AdamW(0.002, 1e-5)
> AdaBound(0.002, 1e-8)
>
>
> ———————————————-
>
> “Do any of these conclusions change when trying out a very small or very large batch size?”
>
> Although an interesting question, it was out of scope with respect to our initial goal of evaluating whether adaptive methods can outperform SGD using the recommended hyperparameters i.e. the batch sizes used in Zagoruyko & Komodakis, Merity et al., and He et al. Unfortunately we cannot re-run our experiments by the rebuttal deadline, but we do believe batch size would be interesting to investigate.  For the camera-ready version, we will generate heatmaps, like the ones in Figure 2, that explore batch size as a hyperparameter.

---

### Author Response · Authors · 2019-11-13
**Comments on revision and concurrent submission**

We thank the reviewers.

We would like to emphasize that our main contribution is not Delayed Adam and its convergence analysis, but our newly-proposed optimizer, AvaGrad.  Additionally, we establish the empirical finding that adaptive methods can outperform SGD across different tasks/datasets, when training distinct architectures -- even in settings where SGD has been universally adopted, such as image classification on ImageNet. These findings require proper tuning of \epsilon, whose optimal values are as large as \epsilon=10, a value 9 orders of magnitude larger than the recommended. Without the decoupling between the learning rate and \epsilon offered by AvaGrad, proper tuning can be extremely costly: in total, we performed over 450 runs to assess the validation performance with different settings for \alpha and \epsilon for each adaptive method.

We have also revised the paper, incorporating changes suggested in the reviews:

   - Our experimental setup now includes AdaShift, following the same protocol we used for other adaptive methods. CIFAR and PTB results are shown in Table 1; ImageNet results will be added to the camera-ready version.

   - We re-tuned SGD on the large-scale experiments (Wide ResNet 28-10 on CIFAR, 4x1000 LSTM on Penn Treebank) to make the comparison against adaptive methods stronger. In all cases, the learning rate that performed the best was the same one chosen from the previous protocol (the one that performed best in the small-scale experiment), hence the numbers are the same.


Reviewers may want to take note of the following concurrent ICLR 2020 submission and the discussion surrounding it:

On Empirical Comparisons of Optimizers for Deep Learning
https://openreview.net/forum?id=HygrAR4tPS

It makes similar observations about the superiority of existing adaptive optimizers when hyperparameters are properly tuned, echoing our results on this matter.  Our contributions, however, go beyond this mere observation, as we also propose, theoretically motivate, and experimentally evaluate a better adaptive optimizer: AvaGrad.

Finally, we are preparing a code repository that we will make publicly available in the near future.

---

### Decision · Program_Chairs · 2019-12-19

**Decision:**

Reject

**Comment:**

This paper proposes an adaptive gradient method for optimization in deep learning called AvaGrad.  The authors argue that AvaGrad greatly simplifies hyperparameter search (over e.g. ADAM) and demonstrate competitive performance on benchmark image and text problems.  In thorough reviews, thorough author response and discussion by the reviewers (which are are all appreciated) a few concerns about the work came to light and were debated.  One reviewer was compelled by the author response to raise their recommendation to weak accept.  However, none of the reviewers felt strongly enough to champion the paper for acceptance and even the reviewer assigning the highest score had reservations.  A major issue of debate was the treatment of hyperparameters, i.e. that the authors tuned hyperparameters on a smaller problem and then assumed these would extrapolate to larger problems. In a largely empirical paper this does seem to be a significant concern.  The space of adaptive optimizers for deep learning is a crowded one and thus the empirical (or theoretical) burden of proof of superiority is high.  The authors state regarding a concurrent submission: "when hyperparameters are properly tuned, echoing our results on this matter", however, it seems that the reviewers disagree that the hyperparameters are indeed properly tuned in this paper.  It's due to these remaining reservations that the recommendation is to reject.